# Smartphone dependence, addiction, and insomnia among medical students during the COVID-19 pandemic

Virgilio E. Failoc-Rojas[1,2], Alicia Torres-Mera[3], Darwin A. León-Figueroa [4,5], David Lira[6], Flor M. Santander-Hernández[7], Miguel A. Guevara-Morales[7], Cristian Díaz-Vélez[8], Mario J. Valladares-Garrido [9]*

1 Vicerrectorado de Investigación, Universidad San Ignacio de Loyola, Lima, Peru, 2 Análisis estadísticos, MedStat-Educación e Investigación, Lima, Peru, 3 Facultad de Medicina, Universidad Nacional Pedro Ruiz Gallo, Lambayeque, Peru, 4 Facultad de Medicina, Universidad de San Martin de Porres, Chiclayo, Peru, 5 Hospital Nacional Sergio E. Bernales, Lima, Perú, 6 Servicio de Neurología, Instituto Peruano de Neurociencias, Lima, Perú, 7 Universidad Cesar Vallejo, Piura, Peru, 8 Instituto Científico de investigación, Universidad Andina del Cusco, Cusco, Peru, 9 Escuela de Medicina Humana, Universidad Señor de Sipán, Chiclayo, Peru

* vgarrido@uss.edu.pe

## Abstract

### Introduction

During the COVID-19 pandemic, smartphone use among students increased, even before going to sleep, which resulted in an increased risk of dependence, addiction, and sleep problems such as insomnia and daytime somnolence. The objective of this study was to determine the association between different levels of problematic smartphone use (dependence and addiction) and insomnia among medical students. In this study, "dependence" refers to subclinical, yet compulsive, smartphone use, while "addiction" denotes a more severe behavioral disruption involving loss of control and functional impairment.

### Materials and methods

This was a secondary cross-sectional study of a cohort of medical students from Piura (Peru) during the COVID-19 pandemic in 2020. The study included 370 participants. Insomnia was assessed using the Insomnia Severity Index, and the extent of smartphone use was determined using the Smartphone Dependence and Addiction Scale. A chi-squared test was used for the bivariate analysis, and prevalence ratios (PR) were estimated using Poisson regression with crude and adjusted models and 95% confidence intervals (CI).

### Results

Insomnia was reported by 67.8% of participants. Smartphone dependence was identified in 67.8% of students, and 10.8% met the criteria for addiction. The prevalence

**Data availability statement:** All relevant data are within the paper and its Supporting Information files.

**Funding:** This research is self-financed. MJV-G was supported by the Fogarty International Center of the National Institutes of Mental Health (NIMH) under Award Number D43TW009343 and the University of California Global Health Institute (UCGHI).

**Competing interests:** The authors have declared that no competing interests exist.

of insomnia was notably higher among students aged ≥26 years (80.0%) and among those with symptoms of depression (79.3%) and anxiety (81.9%). After adjusting for potential confounders, students classified as dependent were 1.43 times more likely to report insomnia (aPR: 1.43; 95% CI: 1.10–1.86), while those classified as addicted showed an even higher association (aPR: 1.66; 95% CI: 1.26–2.17), compared to nondependent users.

## Conclusion

Both smartphone dependence and addiction were significantly associated with a higher prevalence of insomnia among medical students. These findings highlight the need for preventive strategies targeting problematic smartphone use to promote better sleep health in academic settings.

---

## Introduction

Over the last decade, smartphones have become a means of communication, entertainment, and productivity, thus becoming indispensable in our daily lives [1]. More than 5 billion (63.5%) people use the internet worldwide, and 92% of them access the internet from a smartphone [2]. As of 2021, approximately 89% of the Peruvian population aged ≥6 years has used a mobile phone, and 88.5% access the internet through a smartphone [3].

During the COVID-19 pandemic, quarantine was one of the effective measures used by Peru and other states to control the spread of the virus [4]. These restrictive measures could have increased some psychological disorders such as depression, anxiety, anguish, and suicidal thoughts, as well as several sleep problems such as insomnia [5,6]. With the closure of educational institutions, e-learning was developed at a national level, and students used computers, tablets, smartphones, or other devices to access e-learning materials [7,8]. Smartphone use increased by up to 57.2% among university students during the quarantine [9].

Because of the changes in the learning method and their lifestyle, medical students showed an increase in the use of smartphones and the internet, even at night; this increased the risk of addiction or dependence among these students [10]. Previous studies have shown that over 50% of medical students may exhibit smartphone addiction [10,11].

Excessive use of smartphones could generate the need to constantly check the news, stay connected with friends, 'not miss anything,' and seek approval from others, leading to constant restlessness, feelings of loneliness, and low self-esteem [12,13]. In addition, excessive use of smartphones before going to sleep has been associated with poor sleep quality, less rest, worsening insomnia, and daytime somnolence [14]; this adversely affects the quality of life [15]. As a consequence, up to 43% of students often experience a lack of energy, and 30% may develop unhealthy habits, such as less exercise, eating fast food, and weight gain. Moreover, their academic performance may decrease by up to 25% [16]. All of the above has

been associated with stress (OR = 2.39; p < 0.003), anxiety symptoms (OR = 2.04; p < 0.001), and depression (OR = 2.29; p < 0.001) [17].

Several studies have reported that excessive use of smartphones can affect physical and mental health, primarily associated with musculoskeletal pain and insomnia [18]. Long hours of smartphone use lead to poor posture, which can result in cervical spine injuries and neck pain [18,19], as well as carpal tunnel syndrome [20]. In addition, individuals with smartphone addiction are exposed to a higher risk of accidents (OR = 1.9, 95% CI: 1.26–2.86), such as falls, slips, or collisions, due to being engrossed in smartphone use and not paying attention to their surroundings when performing activities like crossing the street or using the subway [21].

Medicine is one of the most stressful fields of study in university education [22,23]. Thus, the health of medical students is a matter of interest. There have been reports of a relationship between smartphone addiction and sleep quality [24]. Although the association between smartphone dependence and addiction and insomnia was reported before [25] and during the COVID-19 pandemic [11,26], there is little conclusive evidence in medical students, particularly at the Latin American level [27]. In our study, we adopted the framework of the Smartphone Dependence and Addiction Scale (SDAS), which categorizes problematic smartphone use into two increasing severity levels: dependence and addiction. Dependence involves compulsive but still manageable usage, whereas addiction reflects a more intense behavioral disruption marked by impaired control and interference with daily functioning [28]. Previous studies have been conducted with a small number of participants [25]; furthermore, they used instruments to assess only the quality of sleep and changes thereof [11,26], thus preventing the assessment of insomnia among participants and the use of statistical methods to determine a better association.

Moreover, other studies have not measured mental health variables such as depression and anxiety [11,26], which are associated with insomnia and smartphone use [29,30]. In this study, we determined whether smartphone dependence and addiction were associated with insomnia among medical students during the COVID-19 pandemic and whether they were influenced by other variables such as depression, anxiety, and nutritional status. Therefore, we will provide evidence that can help develop interventions to regulate smartphone use, aimed at improving students' sleep quality and reducing the impact on their academic life.

## Materials and methods

### Study design and setting

We conducted a cross-sectional analytical study based on secondary data among medical students in Piura, Peru, during the COVID-19 pandemic [30]. This research is a secondary analysis of data originally collected for a previously published study on smartphone overuse, depression, and anxiety (Santander-Hernández et al., 2022). Unlike the original publication, the present work addresses a distinct research question by evaluating smartphone dependence and addiction—as defined by SDAS cutoffs—as the main exposure and insomnia as the primary outcome. This cross-sectional survey was carried out from July to October 2020, coinciding with Peru's first wave of the pandemic and the implementation of strict social distancing measures, including restrictions on public spaces and limited walking hours. Universities fully transitioned to online learning platforms for medical education, and students attended virtual lectures during the study period.

A convenience sampling strategy was used, contacting the class presidents of each university to estimate the total number of students. Three universities were selected for this study, all of which offer the human medicine program in the Piura region, one of the most affected areas in terms of COVID-19 morbidity and mortality in Peru. These universities were selected based on their availability and willingness to participate, and no specific sampling strategy was used to select them. Surveys were designed using Google Forms and distributed through invitation messages in WhatsApp groups commonly used by students. While this method allowed broad dissemination, it may have affected response rates, which are further discussed in the analysis section. Measures were implemented to avoid duplicate responses, and incomplete surveys were excluded from the analysis.

All information was stored in anonymized databases. After completing the survey, we invited all medical students at the three universities to participate in educational sessions provided by psychiatrists. These sessions were held to thank participants and universities for their support of the investigation and contribute to the prevention of excessive smartphone use among medical students.

### Universe, population, and sample: Selection criteria

The target population included all enrolled medical students, regardless of academic year, from the primary study [30]. The sample included 370 students from three universities, representing 16.6% of the population. Specifically, 151 students were from the first university, 121 from the second, and 98 from the third. The selection criterion was that medical students should have a smartphone with internet access for daily use. We excluded participants who had a doctor's diagnosis of depression or anxiety or who had been treated with antidepressants or anxiolytics in the previous year. A total of 38 participants were excluded due to a history of depression or anxiety, resulting in the final sample size.

### Instruments

**Insomnia Severity Index (ISI):** ISI was developed by Bastien et al. [31] to quantify the severity of perceived insomnia according to the DSM-IV criteria. It is a 7-point self-report instrument rated on a 5-point Likert scale (from 0 = not at all to 4 = extremely). Overall scores range from 0 to 28. The severity of insomnia was classified as clinically unimportant (0–7 points), subthreshold (8–14 points), moderate (15–21 points), or severe (22–28 points). We used a Spanish version [32] of ISI, which was validated in medical students and their social networks. Cronbach's alpha coefficient in the Spanish sample was 0.82 [32], whereas it was 0.87 in the present investigation.

**Smartphone Dependence and Addiction Scale (SDAS):** It is a 40-item self-report instrument rated on a 5-point Likert scale (0 = strongly disagree, 1 = disagree, 2 = neither agree nor disagree, 3 = agree, and 4 = strongly agree). SDAS was developed by Aranda-López et al. [28] to evaluate smartphone use in a specific Spanish-speaking population. The scale is divided into three components: (1) use, abuse, and addiction of the smartphone and its applications (30 items); (2) personality traits (6 items); and (3) expenditure on mobile applications and games (4 items). Overall scores range from 0 to 160, and higher scores indicate a higher dependence on smartphones. In our study, we used the cutoff points proposed by the original developers of the scale: nondependence (mean score ≤ 2.23), dependence (2.24–3.05), and addiction (≥ 3.06) [28]. These thresholds were based on terciles calculated during the original validation study and refer to the mean score per item, not the total score [28]. Thus, they are fixed and independent of our sample distribution. Cronbach's alpha coefficient in the original sample was 0.81, 0.76, and 0.71 for components 1, 2, and 3, respectively [28]. In this investigation, Cronbach's alpha coefficient was 0.93 for the global scale and 0.93, 0.71, and 0.76 for components 1, 2, and 3, respectively.

In this study, smartphone use was classified into three categories: nondependence, dependence, and addiction, based on severity. While dependence reflects problematic but still functional use, addiction denotes a more severe level, marked by loss of control and functional impairment.

**Patient Health Questionnaire-9 (PHQ-9):** The PHQ-9 was developed by Kroenke et al. [33] to identify major depression according to DSM-IV criteria and provides total scores from 0 to 27. It is a 9-item self-report questionnaire using a 4-point Likert scale (0 = never, 1 = several days, 2 = more than half the days, and 3 = almost every day). Symptom severity is categorized as minimal (0–4), mild (5–9), moderate (10–14), moderately severe (15–19), and severe (20–27). For analysis, we dichotomized the variable into two categories: participants with scores of 0–4 were considered as having no depression, and those with scores of 5 or more were categorized as having depressive symptoms. For this study, we used a Colombian version of the PHQ-9 [34], which has been validated in medical students with similar characteristics to our study population and with a Cronbach's alpha coefficient of 0.83, which was 0.91 in our sample.

**The General Anxiety Disorder Scale-7 (GAD-7)** is a 7-item self-report instrument, using a 4-point Likert scale, designed to assess anxiety symptoms. For anxiety, severity of symptoms was categorized as minimal (0–4), mild (5–9),

moderate (10–14), and severe (15–21). Similarly, the anxiety variable was divided into two categories for analysis: participants with scores of 0–4 were considered not to have anxiety symptoms, while those with scores of 5 or more were categorized as having anxiety symptoms. We used the Spanish version of the GAD-7 [35], which has been validated in the general population, with a Cronbach's alpha coefficient of 0.94 in the Spanish sample and 0.93 in this study.

## Variables

**Outcome variable.** In this study, insomnia was considered the outcome variable of interest, which was determined using the Insomnia Severity Index (ISI) and operationally defined as a total score greater than 7 points.

**Variable exposure.** The total score on the Smartphone Dependence and Addiction Scale (SDAS) ranges from 0 to 160, where higher scores reflect greater levels of smartphone use. In this study, participants were classified using fixed cutoff points established in the original validation of the scale, based on the mean score per item: nondependence (≤2.23), dependence (2.24–3.05), and addiction (≥3.06) [28]. These categories reflect increasing severity of problematic smartphone use and are independent of the sample distribution.

In this study, the term "smartphone" referred to mobile devices with internet access and apps for social networking, messaging, education, or entertainment, all of which may influence usage patterns and potential addiction [36]. Smartphone dependence is a behavioral disorder that involves compulsive use of the device, affecting daily activities, relationships, and health. Sufferers feel anxious or irritable when without the phone and experience a constant need to check it. Although they try to control their use, they often fail. This behavior shares similarities with substance use disorders, reflecting profound psychological and social implications [37].

**Secondary independent variables.** Depressive symptoms were defined operationally as a score greater than 4 points, obtained from the sum of responses to the 9 items of the Patient Health Questionnaire-9 (PHQ-9). Participants with a total score of 0–4 were considered not to have depressive symptoms, while those with a score greater than 4 were categorized as having depressive symptoms.

The anxiety variable was defined operationally as a score greater than 4 points, obtained from the sum of responses to the 7 items of the Generalized Anxiety Disorder Scale-7 (GAD-7). Participants with a total score of 0–4 were considered not to have anxiety symptoms, while those with a score greater than 4 were categorized as having anxiety symptoms.

Age was assessed through self-report, and participants provided their exact age, which was then categorized into the following groups: 16–20 years, 21–25 years, and ≥26 years. Sex was also self-reported by the participants. Marital status was recorded based on self-report, with participants indicating whether they were single or in a relationship (e.g., married, cohabiting, or dating). Information on whether a family member had died due to COVID-19 was self-reported by the participants. Nutritional status was measured using self-reported weight and height, from which Body Mass Index (BMI) was calculated. Based on the BMI, participants were categorized into two groups: normal weight and overweight/obese, following standard BMI classifications.

## Data analysis

Statistical analysis was performed using Stata v. 16.1 (S1 Table). Variables of interest were initially described with frequencies (n, %). For analysis, an ISI score >8 points was defined as the presence of insomnia. Therefore, the variables followed a dichotomous distribution (0 = absence, 1 = presence) to facilitate the interpretation for decision-making. Bivariate analyses were performed using chi-squared tests to statistically compare the rate of presence of insomnia by covariates. Generalized linear models with a Poisson distribution, log link function, and robust variance were used to determine the association between insomnia and dependence on and addiction to smartphone use. A multivariate analysis was conducted after adjusting for possible confounding factors (age, sex, marital status, body mass index, family member who died of COVID-19, depression, and anxiety). Prevalence ratios (PR) and 95% confidence intervals (CI) were reported. The significance level was set at 5%. To ensure model convergence and allow the estimation of prevalence ratios (PR), robust

variance was used in the Poisson regression. This approach was employed to account for potential correlations and heteroscedasticity in the data.

## Ethical considerations and participant protection

This study was approved by the Research Ethics Committee of César Vallejo University, Piura, Peru (102P-CEI-EPM-UCV-2020). All participants provided virtual informed consent before proceeding with the survey. This was achieved through a consent form embedded at the beginning of the online questionnaire. Participants were required to read and agree to the consent terms before accessing the survey questions. For participants who were minors, consent was also obtained from their parents or legal guardians prior to their participation. We followed the ethical principles listed in the Declaration of Helsinki.

Data was used exclusively for research purposes and kept confidential. To address potential emotional distress related to sensitive questions, we carefully explained these items to participants before starting the survey. Additionally, the survey was anonymous, and participants were informed that they could withdraw at any time without providing a reason.

Educational sessions were conducted for all participating students after completing the survey, as their responses remained anonymous. At the end of the survey, participants were provided with a link to resources on emotional support, prevention of negative emotional states, and contact information for local mental health centers. Participation was entirely voluntary, and no monetary compensation or academic credit was provided. Responding to the survey had no influence on students' grades, course evaluations, or academic standing.

## Results

Approximately 370 students were included in the study. Most participants were women (61.9%) between 16 and 20 years of age (56.8%). Only a small proportion of participants reported the death of a family member because of COVID-19 (25.4%), and a larger proportion reported experiencing symptoms of depression (78.4%) and anxiety (68.9%). A total of 78.6% of participants were classified as having some level of problematic smartphone use, with 67.8% presenting dependence and 10.8% reaching the threshold for addiction, based on the SDAS scoring system. These categories represent increasing severity, with addiction reflecting a more disruptive pattern of use compared to dependence. Finally, 67.8% of participants reported experiencing symptoms of insomnia classified according to ISI. Table 1 shows the characteristics of the population evaluated in this study.

The prevalence of insomnia among participants in this study varied by age, with the highest percentage of insomnia occurring among participants aged ≥26 years (80.0%). There were no significant differences in the prevalence of insomnia between men (70.9%) and women (65.9%) (p=0.319). The prevalence of insomnia was similar between participants who reported the death of a family member because of COVID-19 (63.8%) and those who did not (69.2%) (p=0.335). However, there were significant differences in the prevalence of insomnia among participants who reported experiencing symptoms of depression (79.3%) and those who did not (26.3%) (p<0.001) and among participants who reported experiencing symptoms of anxiety (81.9%) and those who did not (36.5%) (p<0.001). The prevalence of insomnia was higher among participants who were classified as smartphone-dependent (72.1%) than among those who were not (38.0%) (p<0.001) (Table 2).

The results indicated that the prevalence of insomnia was significantly higher among individuals with problematic smartphone use. Those classified with dependence were associated with a 1.43-fold higher prevalence of insomnia (aPR: 1.43; 95% CI: 1.10–1.86; p=0.008), while those categorized as addicted were associated with a 1.66-fold higher prevalence (aPR: 1.66; 95% CI: 1.26–2.17; p<0.001), compared to nondependent users. This pattern suggests an association gradient, where greater severity of smartphone use corresponds to a higher likelihood of insomnia. The results also indicated that sex (female), normal nutritional status, and being in a relationship were not significantly associated with insomnia. Additionally, there were no statistically significant differences in insomnia prevalence between individuals aged 21–25 years or ≥26 years compared to those aged 16–20 years (Table 3).

**Table 1. Description of medical students during the COVID-19 pandemic (n = 370).**

| Characteristics | Frequency | Percentage |
|---|---|---|
| **Age (years)** | | |
| 16–20 | 210 | 56.8 |
| 21–25 | 135 | 36.5 |
| ≥ 26 | 25 | 6.8 |
| **Sex** | | |
| Male | 141 | 38.1 |
| Female | 229 | 61.9 |
| **Marital status** | | |
| Married or in a relationship | 363 | 98.1 |
| Single | 7 | 1.9 |
| **Family member deceased because of COVID-19** | | |
| Yes | 94 | 25.4 |
| No | 276 | 74.6 |
| **Depression** | | |
| Yes | 290 | 78.4 |
| No | 80 | 21.6 |
| **Anxiety** | | |
| Yes | 255 | 68.9 |
| No | 115 | 31.1 |
| **Nutritional status** | | |
| Normal | 242 | 65.4 |
| Overweight/Obese | 128 | 34.6 |
| **Insomnia** | | |
| Yes | 251 | 67.8 |
| No | 119 | 32.2 |
| **Dependence on smartphones** | | |
| No | 79 | 21.4 |
| Dependent | 251 | 67.8 |
| Addiction | 40 | 10.8 |

## Discussion

### Prevalence of insomnia among medical students

Of the students, 67.8% reported insomnia during the first wave of the COVID-19 pandemic. However, this differs from that reported in Saudi Arabia by Alrashed et al. [38]. In 2021, they reported a prevalence of insomnia of 34.9% among medical students during the COVID-19 pandemic [38]. In 2022, a study in Saudi Arabia reported that 41% of students had a prevalence of moderate to severe insomnia during the COVID-19 pandemic [39]. A multinational study conducted in 2021 during the COVID-19 pandemic by Tahir et al. reported that 73.5% of medical students from the Dominican Republic, Egypt, Guyana, India, Mexico, Pakistan, and Sudan had insomnia [40]. Another study conducted by Pallares et al. in Colombia during the COVID-19 pandemic reported that 60.2% of students from the School of Health Sciences had symptoms of insomnia [41]. In 2023, Allende-Rayme et al. reported that, during the COVID-19 pandemic, 90.48% of medical students from a Peruvian university had insomnia [42]. Other results were reported in 2021. Olarte-Durand et al. concluded that 83.9% of medical students at a private university in Peru had insomnia [43]. This could be explained by the fact that confinement, quarantine, and mobility limitations because of COVID-19 had immediate and inevitable negative

**Table 2. Factors associated with insomnia among medical students during the COVID-19 pandemic (n = 370).**

| Characteristics | Insomnia (ISI) | | | | χ² | p |
|---|---|---|---|---|---|---|
| | Yes | | No | | | |
| | Frequency | Percentage | Frequency | Percentage | | |
| **Age (years)** | | | | | 1.926 | 0.382 |
| 16–20 | 142 | 67.8 | 68 | 32.2 | | |
| 21–25 | 89 | 65.9 | 46 | 34.1 | | |
| ≥ 26 | 20 | 80.0 | 5 | 20.0 | | |
| **Sex** | | | | | 0.993 | 0.319 |
| Male | 100 | 70.9 | 41 | 29.1 | | |
| Female | 151 | 65.9 | 78 | 34.1 | | |
| **Family member deceased because of COVID-19** | | | | | 0.928 | 0.335 |
| Yes | 60 | 63.8 | 34 | 36.2 | | |
| No | 191 | 69.2 | 85 | 30.8 | | |
| **Depression** | | | | | 80.911 | **<0.001** |
| Yes | 230 | 79.3 | 60 | 20.7 | | |
| No | 21 | 26.3 | 59 | 73.7 | | |
| **Anxiety** | | | | | 75.002 | **<0.001** |
| Yes | 209 | 81.9 | 46 | 18.1 | | |
| No | 42 | 36.5 | 73 | 63.5 | | |
| **Nutritional status** | | | | | 0.951 | 0.329 |
| Normal | 160 | 66.1 | 82 | 33.9 | | |
| Overweight/obese | 91 | 71.1 | 37 | 28.9 | | |
| **Marital status** | | | | | 0.042 | 0.837 |
| Married or in a relationship | 246 | 67.8 | 117 | 32.2 | | |
| Single | 5 | 71.4 | 2 | 28.6 | | |
| **Dependence on smartphones** | | | | | 53.356 | **<0.001** |
| No | 30 | 38.0 | 49 | 62.0 | | |
| Dependent | 181 | 72.1 | 70 | 27.9 | | |
| Addiction | 40 | 100.0 | 0 | 0.0 | | |

χ², Chi-squared test

ISI: Insomnia Severity Index

effects that significantly altered daily schedules and routines. This led to a virtual adaptation of education, leading to an increased use of smartphones and changes in sleep habits and patterns (in terms of wake-up times, bedtimes, and sleep hours), which are connected to circadian rhythms that regulate sleep; this subjected medical students to insufficient sleep [39,44]. Smartphones emit blue light, which can suppress the production of melatonin—a hormone that helps regulate sleep and circadian cycles [45]. Exposure to blue light at night can delay sleep onset and alter sleep quality, which can contribute to insomnia [46].

## Association between smartphone dependence and addiction and insomnia

Smartphone dependence was associated with a 43% higher prevalence of insomnia, while smartphone addiction showed an even stronger association, with a 66% higher prevalence. These findings reflect a severity gradient in which addiction represents a more advanced stage of problematic smartphone use compared to dependence, as categorized by the SDAS.

 

**Table 3. Crude and adjusted association between smartphone use and insomnia among medical students during the COVID-19 pandemic (n = 370).**

| Characteristics | Simple analysis | | | Adjusted analysis | | |
|---|---|---|---|---|---|---|
| | cPR | 95% CI | P | aPR | 95% CI | p |
| **Age (years)** | | | | | | |
| 16–20 | Ref. | | | Ref. | | |
| 21–25 | 0.97 | 0.84–1.14 | 0.746 | 0.98 | 0.85–1.13 | 0.739 |
| ≥26 | 1.18 | 0.95–1.47 | 0.130 | 1.09 | 0.89–1.33 | 0.400 |
| **Sex** | | | | | | |
| Male | Ref. | | | Ref. | | |
| Female | 0.93 | 0.81–1.07 | 0.311 | 0.95 | 0.84–1.07 | 0.739 |
| **Family member deceased because of COVID-19** | | | | | | |
| No | Ref. | | | Ref. | | |
| Yes | 0.92 | 0.78–1.09 | 0.356 | 0.97 | 0.83–1.13 | 0.679 |
| **Depression** | | | | | | |
| No | Ref. | | | Ref. | | |
| Yes | 3.02 | 2.08–4.38 | **<0.001** | 2.10 | 1.41–3.12 | **<0.001** |
| **Anxiety** | | | | | | |
| No | Ref. | | | Ref. | | |
| Yes | 2.24 | 1.75–2.88 | **<0.001** | 1.48 | 1.15–1.90 | **0.002** |
| **Nutritional status** | | | | | | |
| Normal | Ref. | | | Ref. | | |
| Overweight/obese | 1.08 | 0.93–1.24 | 0.319 | 0.97 | 0.83–1.13 | 0.679 |
| **Marital status** | | | | | | |
| Married or in a relationship | Ref. | | | Ref. | | |
| Single | 0.95 | 0.59–1.52 | 0.828 | 0.94 | 0.83–1.13 | 0.679 |
| **Dependence on smartphones** | | | | | | |
| No | Ref. | | | Ref. | | |
| Dependent | 1.9 | 1.42–2.54 | **<0.001** | 1.43 | 1.10–1.86 | **0.008** |
| Addiction | 2.63 | 1.98–3.49 | **<0.001** | 1.66 | 1.26–2.17 | **<0.001** |

cPR, Crude Prevalence Ratio; aPR, Adjusted Prevalence Ratio.

p-values obtained via generalized linear models with Poisson family and robust variance. 95% CI: 95% confidence interval.

Note: Categories are based on fixed mean score cutoffs from the original SDAS validation: nondependence (≤2.23), dependence (2.24–3.05), and addiction (≥3.06).

In 2022, similar results were reported by Liu et al., who found a strong association between smartphone dependence and insomnia (r = 0.566) among medical students from China during the COVID-19 pandemic [10]. In 2021, a study conducted in Indonesia by Indrakusuma et al. also found that smartphone dependence among medical students had a weak positive correlation (r = 0.162) with the prevalence of insomnia during the COVID-19 pandemic [47]. In 2021, during the COVID-19 pandemic in India, Telgote et al. reported a significant correlation (r = 0.35) between smartphone addiction and insomnia severity among medical students [48]. In 2021, Alageel et al. conducted a prepandemic study and reported smartphone dependence among graduate students, who were approximately twice as likely to experience insomnia (odds ratio [OR] = 2.113) [49]. This association could be explained by the increase in the use of smartphones owing to the development of virtual education during the COVID-19 pandemic. This leads to medical students spending more time searching for information on the internet and having unhealthy sleep habits [50]. Thus, prolonged use of smartphones negatively affects the body's circadian rhythm, which results in insomnia [51].

These findings reinforce existing literature on the relationship between smartphone dependence and insomnia among medical students. However, our study adds value by focusing on a Latin American population, where such associations remain underexplored. Unlike previous studies conducted primarily in Asia and the Middle East [10,47–49], our research captures the unique academic and socioeconomic stressors experienced by medical students in Peru during the COVID-19 pandemic. Additionally, by quantifying the increased prevalence of insomnia associated with smartphone dependence, we provide empirical evidence that supports the urgent need for interventions targeting digital health literacy and responsible smartphone use among future healthcare professionals.

It is important to note that the categorization of smartphone use levels was based on fixed thresholds established in the original validation of the SDAS, not on sample-derived percentiles. This methodological choice ensures comparability with other studies using the same instrument and strengthens the external validity of our findings.

## Other factors associated with insomnia

In our study, depression was associated with a higher prevalence of insomnia. This is like what was reported in China in 2023 by Zhang et al., who showed a higher prevalence of insomnia among medical students with depression (OR = 2.12) [52]. In 2021, Choi et al. reported in the Republic of Korea that the prevalence of insomnia symptoms was significantly higher among participants with symptoms of depression than in those without said symptoms (64.7% vs. 8.3%, respectively; $p < 0.001$) [53]. In 2021, Liu et al. showed that depression among Chinese medical students was positively correlated with insomnia ($r = 0.679$, $p < 0.001$) [54]. In 2022, Copaja-Corzo et al. reported a significant association between symptoms of insomnia and depression among Peruvian medical students (adjusted PR [aPR] = 2.03) [55]. In Saudi Arabia, Alrashed et al. reported in 2021 that medical students with severe depression had a higher probability of reporting experiencing symptoms of insomnia (OR = 3.10) [38]. Medical students are at a high risk of developing depression symptoms because of the stress associated with their education, including psychological and academic factors [56,57]. This leads to changes in their biological clock and alterations in the production of sleep-regulating hormones, such as melatonin; consequently, the ability to fall asleep and maintain sleep can become impaired, thus causing insomnia [57].

Anxiety was associated with a higher prevalence of insomnia among the population in our study. This is like what was reported in Norway by Haugland et al. [58]. In 2021, they concluded that students with anxiety symptoms were at a higher risk of developing insomnia (RR = 1.03) [58]. In 2023, Zhang et al. reported in China that the probability of developing insomnia increased among medical students with anxiety (OR = 2.35) [52]. In Pakistan, Mahmood-ul-Hassan et al. reported in 2021 that medical and nonmedical students showed a statistically significant correlation between anxiety scores and insomnia scores ($r = 0.742$) [59]. In Taiwan, Chen et al. reported in 2022 that medical students with anxiety had more symptoms of insomnia ($\beta = 0.51$, $p < 0.001$) [60]. In Peru, Copaja-Corzo et al. reported in 2021 that the prevalence of insomnia increased by 48% among medical students with anxiety (aPR = 1.48) [55]. In 2022, Olarte-Durand et al. reported that male Peruvian medical students with anxiety had increased symptoms of insomnia (aPR = 1.34) [43]. This relationship could be explained by diverse factors associated with the academic environment and lifestyle of medical students [61]. The accumulation of academic hours, intense study demands, and exposure to emotionally challenging courses can lead to considerable levels of anxiety among students. This can have negative effects, such as sleep deprivation and disruption of the circadian rhythm, which, in turn, can lead to insomnia [62].

## Implications of the findings in medical education

Because of the COVID-19 pandemic, rigorous control measures, such as social distancing, were implemented worldwide. This situation caused an adaptation of online medical education, thus leading to an addiction to smartphones among medical students [26,57,58,62]. Smartphone use has become increasingly common and has transformed the way we interact and communicate. However, smartphone addiction is an emerging public health issue with several causes contributing to

its prevalence, including social, psychological, technological, and familial aspects [63]. Nevertheless, the COVID-19 pandemic was the most crucial factor that influenced the level of smartphone addiction [64].

Exposure to artificial light from smartphones and other electronic devices before going to sleep can interfere with the body's natural circadian rhythm, making it more difficult to fall asleep and affecting sleep quality [65]. This can lead to a decline in cognitive and physical performance, thus affecting people's ability to work and perform daily activities effectively [66].

Beyond individual health consequences, the high prevalence of insomnia among medical students due to excessive smartphone use raises concerns about its long-term impact on academic performance, cognitive function, and professional readiness. Sleep deprivation can impair learning, decision-making, and clinical reasoning skills, all of which are crucial in medical education and future clinical practice [67]. Addressing these issues through institutional policies—such as curriculum-integrated education on sleep hygiene, time management, and digital well-being—could enhance both academic success and long-term professional competency in healthcare settings.

In our investigation, we discovered a direct and significant relationship between smartphone addiction and the presence of insomnia. Moreover, medical students' use of smartphones before going to sleep affected their sleep quality and ability to fall asleep, which could lead to the development of chronic sleep problems. These health issues could negatively affect the quality of life, academic performance, learning ability, and memory of medical students [67].

Considering the findings, it is crucial to expand education on smartphone dependence to all healthcare providers. Screening for smartphone addiction should become a routine practice, particularly when addressing sleep disorders or cognitive decline, as early identification can help prevent the negative impacts on sleep quality, academic performance, and overall health. Raising awareness among medical students about the risks of excessive smartphone use, especially before bedtime, can play a pivotal role in mitigating these effects and promoting healthier sleep habits [68,69].

To further support this initiative, integrating education on smartphone addiction and its health implications into medical curricula is essential. Our study contributes by specifically focusing on medical students in Piura, a region in northern Peru, during the COVID-19 pandemic, providing valuable insights into the unique challenges faced by this population. Standardizing this content across medical schools will ensure that future healthcare professionals are well-equipped to recognize and address the issue in both themselves and their patients [70]. By focusing on this under-researched cohort, particularly within the context of remote learning, our study offers a detailed examination of how pandemic-related changes have exacerbated smartphone addiction and related health issues. This comprehensive approach will contribute to better management of technology-related sleep disorders and foster healthier lifestyles for both medical students and the wider population [71].

Additionally, our findings emphasize the need for medical schools to implement proactive strategies, such as digital detox programs and structured sleep hygiene workshops. Given that excessive smartphone use before bedtime is linked to cognitive impairment and reduced academic performance, institutions should consider integrating mandatory well-being modules to help students develop healthier technology habits. Future research should explore the effectiveness of such interventions in reducing insomnia and improving academic outcomes among medical students.

## Limitations and strengths

This study has several limitations. First, the use of an online survey for data collection may have introduced selection bias, as medical students who frequently use social media and the internet may have been more likely to participate. This could potentially overestimate the association between smartphone addiction and insomnia if students with higher technology use were more inclined to report sleep disturbances; conversely, students who experience insomnia but do not engage frequently with online platforms may have been underrepresented, leading to a possible underestimation of the true prevalence.

Second, the findings may not be fully generalizable to the broader medical student population, as the data were collected from a single region of Peru, introducing potential selection bias. The cultural, educational, and socioeconomic factors unique to this setting may limit the extrapolation of our results to other regions or countries. Future studies should consider multi-center designs to enhance external validity. Although we used a convenience sampling strategy and data collection was conducted via WhatsApp groups, the demographic characteristics of our sample—particularly regarding age and sex—are consistent with those reported in recent multicenter mental health studies conducted among Peruvian medical students during the COVID-19 pandemic [72–75]. This comparability suggests that, despite the non-probabilistic nature of our sampling, the findings may still reflect broader patterns within this population. Third, certain variables that could influence both smartphone addiction and sleep patterns were not included in our analysis due to data availability constraints; these included total daily study hours [76], monthly family income [77], physical activity [78], healthy eating habits [79], average sleep duration [77], academic year [49], and academic performance [80]. The omission of these factors may introduce residual confounding, and future research should integrate a more comprehensive set of predictors. Additionally, further studies should explore the frequency, duration, and type of social media applications used [81], the impact of online gaming [82], the presence of suicidal thoughts [83], chronic illnesses [84], and smartphone use during different times of the day [55].

Fourth, our sample consisted exclusively of medical students, which limits the ability to compare smartphone addiction and insomnia across different academic disciplines. Given that study demands and stress levels may vary significantly between fields, comparative research including students from other disciplines is warranted to determine whether these associations are specific to medical students or represent a broader phenomenon [50].

Despite these limitations, our study provides important contributions to literature. To our knowledge, this is the first study from Latin America to examine the association between smartphone dependence and insomnia among medical students, particularly in the context of the COVID-19 pandemic. We used validated instruments to assess key psychological and behavioral factors, ensuring robust data quality. Furthermore, our analysis was conducted using rigorous biostatistical methods to minimize bias and enhance internal validity.

### Relevance of mental health findings

Our findings lay the foundation for the future in this field. Potential areas for further investigation include (1) longitudinal studies assessing the progression of smartphone dependence and addiction and insomnia among medical students; (2) comparative studies evaluating differences between medical students and other university populations; (3) research exploring the effect of smartphone use on sleep quality by differentiating users and non-users; (4) analyses examining the relationship between smartphone use, addiction, and insomnia in relation to academic performance; and (5) studies investigating the interplay between physical activity, smartphone addiction, and sleep disturbances.

By addressing these gaps, future research can contribute to the development of targeted interventions aimed at mitigating the adverse effects of excessive smartphone use on sleep and mental health among medical students. Moreover, our study highlights the importance of understanding smartphone dependence not only as an individual behavioral issue but also as a broader academic and institutional concern. Given this context, structured interventions—such as digital well-being education, sleep hygiene programs, or cognitive-behavioral strategies—should be evaluated for their effectiveness in mitigating insomnia and promoting mental health among medical students.

### Conclusions

In this cross-sectional study, we found that both smartphone dependence and addiction were significantly associated with a higher prevalence of insomnia among medical students during the COVID-19 pandemic. Additionally, those with anxiety and depression were more likely to report severe insomnia symptoms. These results highlight the importance of raising awareness about the potential risks associated with excessive smartphone use, particularly its association with reduced

sleep quality. While causal relationships cannot be established, interventions that address problematic smartphone use may contribute to promoting better sleep hygiene and improving overall well-being.

Our study underscores the pressing need for targeted interventions addressing smartphone dependence among medical students. Medical schools should consider implementing evidence-based guidelines for digital device usage, particularly before bedtime, to promote healthier sleep patterns. Given the well-established link between sleep deprivation and academic underperformance, integrating sleep education and digital detox strategies into medical curricula could help safeguard students' cognitive and professional development.

## Supporting information

**S1 Table. Database.**
(DTA)

**S1 Checklist. STROBE Statement—checklist of items that should be included in reports of observational studies.**
(PDF)

## Author contributions

**Conceptualization:** Virgilio E. Failoc-Rojas, Alicia Torres-Mera, Darwin A. León-Figueroa, David Lira, Flor M. Santander-Hernández, Miguel A. Guevara-Morales, Cristian Díaz-Vélez, Mario Valladares-Garrido.

**Data curation:** Virgilio E. Failoc-Rojas, Alicia Torres-Mera, Darwin A. León-Figueroa, David Lira, Flor M. Santander-Hernández, Miguel A. Guevara-Morales, Cristian Díaz-Vélez, Mario Valladares-Garrido.

**Formal analysis:** Virgilio E. Failoc-Rojas, Darwin A. León-Figueroa, David Lira, Mario Valladares-Garrido.

**Investigation:** Virgilio E. Failoc-Rojas, Alicia Torres-Mera, Darwin A. León-Figueroa, David Lira, Flor M. Santander-Hernández, Miguel A. Guevara-Morales, Cristian Díaz-Vélez, Mario Valladares-Garrido.

**Methodology:** Virgilio E. Failoc-Rojas, Alicia Torres-Mera, Darwin A. León-Figueroa, Flor M. Santander-Hernández, Miguel A. Guevara-Morales, Cristian Díaz-Vélez, Mario Valladares-Garrido.

**Project administration:** Virgilio E. Failoc-Rojas, Darwin A. León-Figueroa, David Lira, Flor M. Santander-Hernández, Miguel A. Guevara-Morales, Mario Valladares-Garrido.

**Resources:** Virgilio E. Failoc-Rojas, Alicia Torres-Mera, David Lira, Flor M. Santander-Hernández, Cristian Díaz-Vélez, Mario Valladares-Garrido.

**Software:** Virgilio E. Failoc-Rojas, Alicia Torres-Mera, Darwin A. León-Figueroa, Flor M. Santander-Hernández, Miguel A. Guevara-Morales, Mario Valladares-Garrido.

**Supervision:** Virgilio E. Failoc-Rojas, Alicia Torres-Mera, Darwin A. León-Figueroa, David Lira, Flor M. Santander-Hernández, Miguel A. Guevara-Morales, Cristian Díaz-Vélez, Mario Valladares-Garrido.

**Validation:** Virgilio E. Failoc-Rojas, Alicia Torres-Mera, Darwin A. León-Figueroa, David Lira, Flor M. Santander-Hernández, Miguel A. Guevara-Morales, Cristian Díaz-Vélez, Mario Valladares-Garrido.

**Visualization:** Virgilio E. Failoc-Rojas, Alicia Torres-Mera, Darwin A. León-Figueroa, David Lira, Flor M. Santander-Hernández, Miguel A. Guevara-Morales, Cristian Díaz-Vélez.

**Writing – original draft:** Virgilio E. Failoc-Rojas, Alicia Torres-Mera, Darwin A. León-Figueroa, David Lira, Flor M. Santander-Hernández, Miguel A. Guevara-Morales, Cristian Díaz-Vélez, Mario Valladares-Garrido.

**Writing – review & editing:** Virgilio E. Failoc-Rojas, Alicia Torres-Mera, Darwin A. León-Figueroa, David Lira, Flor M. Santander-Hernández, Miguel A. Guevara-Morales, Cristian Díaz-Vélez, Mario Valladares-Garrido.

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
