## [Decision Letter · Decision Letter 0]

Dear Dr. Valladares-Garrido,

Thank you for submitting your manuscript to PLOS ONE. After careful consideration, we feel that it has merit but does not fully meet PLOS ONE’s publication criteria as it currently stands. Therefore, we invite you to submit a revised version of the manuscript that addresses the points raised during the review process.

**ACADEMIC EDITOR:**

Thank you for submitting your manuscript to PLOS ONE. The study addresses a highly relevant topic by exploring smartphone dependence, insomnia, and their association with mental health and behavioral patterns, particularly in the context of medical students during the COVID-19 pandemic. The research design is sound, and the findings provide meaningful insights that contribute to the growing body of literature in this field.

After carefully considering the feedback provided by the reviewers, it is evident that while the comments individually suggest minor adjustments, their cumulative scope indicates the need for a more thorough revision. This decision is not a reflection of any fundamental issues with your work but rather an opportunity to ensure that the manuscript meets the highest standards of clarity, methodological transparency, and interpretative rigor.

We suggest expanding the introduction to include additional literature on behavioral addiction and its relationship with mental health conditions. This will help contextualize the study within the broader research landscape and better articulate the research gap your study aims to address. Highlighting the novelty of your approach and findings will strengthen the manuscript further.

In the methods section, please provide greater detail on the sampling strategy, including the number of universities selected, and elaborate on the exclusion process that led to the final sample size. It would also be helpful to clarify how confounding variables, such as nutritional status, were identified and measured, as well as how the adjustments were applied during the analysis. Additionally, we recommend reorganizing the section to separate the description of scales and instruments from operational definitions, which could be included as a distinct subsection for improved readability.

The ethical considerations require clarification, specifically regarding the type of consent obtained from participants. This information should be included explicitly to align with ethical guidelines and ensure transparency.

In the results section, address the discrepancies between the methods and results regarding the categorization of variables such as depression and anxiety. Additionally, refine the interpretation of prevalence ratios to provide a clearer understanding of the findings.

Finally, the discussion would benefit from a deeper reflection on the implications of your findings in addressing the research gap and contributing to future work in this area. Explicit acknowledgment of the methodological limitations, including potential selection biases and the rationale for excluding certain clinical conditions, will enhance the manuscript’s rigor and balance.

We recognize the effort that has already gone into this study and appreciate its potential to contribute meaningfully to the field. We look forward to receiving your revised manuscript and are confident that these changes will further strengthen the work.

We look forward to receiving your revised manuscript.

Kind regards,

Juan Luis Castillo-Navarrete, Ph.D.

Academic Editor

PLOS ONE

Journal Requirements:

Reviewers' comments:

Reviewer's Responses to Questions

**Comments to the Author**

1. Is the manuscript technically sound, and do the data support the conclusions?

Reviewer #1: Yes

Reviewer #2: Yes

Reviewer #3: Yes

2. Has the statistical analysis been performed appropriately and rigorously?

Reviewer #1: Yes

Reviewer #2: Yes

Reviewer #3: Yes

3. Have the authors made all data underlying the findings in their manuscript fully available?

Reviewer #1: Yes

Reviewer #2: Yes

Reviewer #3: Yes

4. Is the manuscript presented in an intelligible fashion and written in standard English?

Reviewer #1: Yes

Reviewer #2: Yes

Reviewer #3: Yes

Reviewer #1: Please read the attached annotated manuscript for comments. The manuscript details findings on associations between various count data variables, namely, smartphone dependence and insomnia and associated sociodemographic variables. Whilst there is nothing inherently problematic in the manuscript, the amount of new information to the area is adequate. Self-reported student stress and smartphone time consumption are significantly associated but this finding is not novel nor unexecpted. Nonetheless, the research design and statistical analyses is correct and the authors do not move beyond what the data suggests.

Reviewer #2: In the introduction section, authors have mentioned limited literature related to the study, they are suggested to add more on behavioral addiction and smart phone addition and its association with mental health conditions. In method section (line no. 97-100), authors mentioned “we acknowledge the inherent limitations of this method, including the potential for selection bias, which may limit the generalizability of our findings. A discussion of these limitations is provided in the results and discussion sections.” This should only be mentioned in the limitation section of the study.

Line 114 and 115, authors have mentioned that “they excluded people who had a doctor's diagnosis of depression, or who had been treated with antidepressants in the previous year. How about anxiety and other clinical conditions, why only excluding depression. As we know that during pandemic more anxiety was reported then depression, kindly clarify this.

Authors are requested to give description of scales/instruments in subsection separately. They are requested to mention the operational definitions separately at the end of method section, it should not be mixed up with the description of scales/instruments.

Reviewer #3: 1. In Ethics statement, please "indicate the form of consent obtained"

2. Please find the comments for given line numbers below:

Line 95. How many universities were selected? Were all universities in Peru included? If not, what sampling strategy was used?

Line 116. How many participants were included in the survey? And how many were excluded to reach the final sample size of 370?

Line 157. I suppose, 15-19 should be categorized as 'moderately severe' instead of 'moderate'.

Line 164. Details on confounding variables could be elaborated. For example, how was the nutritional status measured? Also, were the assumptions of poisson regression met?

Line 175. How were the confounding variables identified, and how was the minimal set of adjusted confounders determine? (e.g. using DAG or any models)

Line 180. I suppose, the participants provided a written consent.

Line 199. In the methods section, it says that depression was a categorical variable. However, in the results, it is presented as a dichotomous variable. Could you explain how it was dichotomized? Same for anxiety, the methods section does not explain how the variable anxiety was treated.

Line 217. Since you are reporting exp beta, an interpretation like this "The prevalence of insomnia was 1.9 times higher among individuals with smartphone dependence compared to those without dependence" may be better than saying "more likely to" (as it sounds more like interpreting likelihood)

Line 223. Since you are talking about the frequency of insomnia, are you referring to table 2?

**Do you want your identity to be public for this peer review?** For information about this choice, including consent withdrawal, please see our Privacy Policy

Reviewer #1: No

Reviewer #2: **Yes: ** Salman Shahzad

Reviewer #3: **Yes: ** Supriya Sherpa

---

## [Author Response · Author response to Decision Letter 1]

11 Feb 2025

Dear Editor,

Thank you very much for reviewing our article, " Smartphone dependence, addiction, and insomnia among medical students during the COVID-19 pandemic ". Your suggestions and comments will be addressed below. Thank you for your valuable time and excellent review.

Editor's comments

1. Editor says: We suggest expanding the introduction to include additional literature on behavioral addiction and its relationship with mental health conditions. This will help contextualize the study within the broader research landscape and better articulate the research gap your study aims to address. Highlighting the novelty of your approach and findings will strengthen the manuscript further.

Our response: Thank you for your comment. As suggested, the introduction has been expanded with literature on smartphone addiction and its association with anxiety and depression.

2. Editor says: In the methods section, please provide greater detail on the sampling strategy, including the number of universities selected, and elaborate on the exclusion process that led to the final sample size. It would also be helpful to clarify how confounding variables, such as nutritional status, were identified and measured, as well as how the adjustments were applied during the analysis. Additionally, we recommend reorganizing the section to separate the description of scales and instruments from operational definitions, which could be included as a distinct subsection for improved readability.

Our response: Thank you for your insightful comments. We appreciate the opportunity to clarify and enhance the methodology section. We have addressed each point as follows:

a) Ethical considerations (form of consent): We have added more detailed information regarding the form of consent obtained, as follows:

"All participants provided virtual informed consent before proceeding with the survey. This was achieved through a consent form embedded at the beginning of the online questionnaire. Participants were required to read and agree to the consent terms before accessing the survey questions.” (lines 398-416).

b) Sampling strategy and universities selected: Three universities were selected for this study, all offering the human medicine program in the Piura region, one of the most affected areas in terms of COVID-19 morbidity and mortality in Peru. The universities were chosen based on availability and willingness to participate, with no specific sampling strategy. The sampling approach was non-probabilistic and based on convenience, given the logistical constraints posed by the pandemic. This is now clarified in lines 201-207.

c) Exclusion criteria (number of participants excluded): The survey initially included a larger number of participants, but 38 were excluded due to a history of depression or anxiety, diagnosed by a healthcare professional or treated with antidepressants/anxiolytics in the previous year. This resulted in a final sample size of 370 participants. This is now included in lines 198-207.

d) Confounding variables (identification and adjustment): Confounding variables were selected based on prior literature, expert opinion, and known relationships with the outcome variables (insomnia and smartphone dependence/addiction). A stepwise approach was used to determine the minimal set of confounders, including age, sex, marital status, BMI, family member deceased due to COVID-19, depression, and anxiety. These variables were tested for inclusion based on statistical significance. This is now clarified in lines 323-382.

e) Separation of scales/instruments and operational definitions: As requested, we have reorganized the methods section to clearly separate the description of scales from the operational definitions. The description of each instrument (Insomnia Severity Index, Smartphone Dependence and Addiction Scale, PHQ-9, and GAD-7) is now in a separate subsection, with operational definitions of the variables of interest placed at the end of the methods section, as suggested.

We hope these changes address your concerns and improve the clarity and quality of the manuscript.

3. Editor says: The ethical considerations require clarification, specifically regarding the type of consent obtained from participants. This information should be included explicitly to align with ethical guidelines and ensure transparency.

Our response: Thank you for your valuable comment. We appreciate your suggestion to clarify the ethical considerations, particularly regarding the type of consent obtained from participants. In response to this, we have expanded the Ethical Considerations section to provide a more comprehensive description of the consent process.

We now specify that virtual informed consent was obtained from all participants before they proceeded with the survey. The consent process was clearly outlined at the beginning of the online questionnaire, where participants were presented with a detailed consent form. This form included comprehensive information about the study's purpose, procedures, potential risks, confidentiality, and their right to withdraw at any time without consequence. Participants were required to read the consent form carefully and electronically agree to the terms by selecting an option that indicated their consent before gaining access to the survey questions. This method ensured that all participants had a clear understanding of the study and voluntarily agreed to participate.

4. Editor says: In the results section, address the discrepancies between the methods and results regarding the categorization of variables such as depression and anxiety. Additionally, refine the interpretation of prevalence ratios to provide a clearer understanding of the findings.

Our response: Thank you for your insightful comment. We apologize for the oversight in the methods section. To clarify, both depression and anxiety were treated as categorical variables based on symptom severity, as follows:

• Depression: Symptoms were categorized into minimal (0-4), mild (5-9), moderate (10-14), moderately severe (15-19), and severe (20-27). For analysis, we dichotomized it into two categories: no depression (0-4) and depressive symptoms (5+).

• Anxiety: Symptoms were categorized into minimal (0-4), mild (5-9), moderate (10-14), and severe (15-21). Similarly, we dichotomized it into no anxiety (0-4) and anxiety symptoms (5+).

We hope this clarification resolves the discrepancies and improves the interpretation of the results.

5. Editor says: Finally, the discussion would benefit from a deeper reflection on the implications of your findings in addressing the research gap and contributing to future work in this area. Explicit acknowledgment of the methodological limitations, including potential selection biases and the rationale for excluding certain clinical conditions, will enhance the manuscript’s rigor and balance.

Our response:

Thank you for your insightful comments and for the opportunity to improve our manuscript. We appreciate your suggestions regarding the discussion section and have made the following revisions:

• We have expanded our discussion on how our findings address the research gap. Specifically, we highlight how our study provides novel insights into smartphone addiction and insomnia among medical students in Latin America, a population that has been underrepresented in previous research.

• We have further elaborated on how our findings can inform future research by suggesting potential studies that explore causal mechanisms, intervention strategies, and comparative analyses across different student populations.

• We have explicitly acknowledged additional methodological limitations, including a more detailed discussion on potential selection bias due to the online survey format and the rationale for excluding certain clinical conditions.

Reviewer #1:

1. Reviewer says: Please read the attached annotated manuscript for comments. The manuscript details findings on associations between various count data variables, namely, smartphone dependence and insomnia and associated sociodemographic variables. Whilst there is nothing inherently problematic in the manuscript, the amount of new information to the area is adequate.

Our response: Thank you for your valuable feedback. We acknowledge that the associations between smartphone dependence, insomnia, and sociodemographic variables, such as student stress, have been well-documented in the literature. However, we believe that our study provides significant value by focusing on medical students during the pandemic, a population that faced unique challenges related to the rapid transition to online learning, increased mental health burdens, and changes in daily routines. The analytical approach we used allowed us to examine these variables in a comprehensive way, providing insight into the specific impact of the pandemic on this group. Our study also contributes to the existing literature by considering the unique challenges faced by medical students in Piura, a region in northern Peru, especially regarding the transition to remote learning during the COVID-19 pandemic. Additionally, we employed a cross-sectional analytical approach, utilizing validated self-report scales to assess smartphone dependence and insomnia, which allows for a more nuanced understanding of these issues in a specific cohort. While the findings may not be novel, we hope to contribute to a better understanding of these issues and their implications for medical education and student well-being during extraordinary circumstances. Furthermore, we have carefully applied statistical analyses to ensure the data is interpreted in the most accurate and meaningful way possible.

Finally, we have emphasized these points in the Implications of Findings section of the manuscript to highlight the unique context of medical students during the pandemic. Moreover, we recognize the limitations of our study in the Discussion section, acknowledging factors such as the cross-sectional design and the self-reported nature of the data, which may limit the generalizability of the findings.

2. Reviewer says: Self-reported student stress and smartphone time consumption are significantly associated but this finding is not novel nor unexecpted. Nonetheless, the research design and statistical analyses is correct, and the authors do not move beyond what the data suggests.

Our response:

We sincerely appreciate the reviewer’s insightful comments, which have helped us refine our manuscript. In response, we have strengthened the discussion by highlighting our study’s contribution to existing knowledge and its broader implications. Specifically, we have expanded the discussion on how our findings add new empirical evidence from a Latin American population and complement previous studies conducted in other regions.

Please find the revised manuscript attached. We hope the changes meet the reviewer’s expectations.

Reviewer #2:

1. Reviewer says: In the introduction section, authors have mentioned limited literature related to the study, they are suggested to add more on behavioral addiction and smart phone addition and its association with mental health conditions.

Our response: Thank you for your comment. As suggested, I have added literature on smartphone addiction and its association to anxiety and depressive symptoms.

In method section (line no. 97-100), authors mentioned “we acknowledge the inherent limitations of this method, including the potential for selection bias, which may limit the generalizability of our findings. A discussion of these limitations is provided in the results and discussion sections.” This should only be mentioned in the limitation section of the study.

Our response: Thank you for your helpful comment. We apologize for the oversight in the methods section. As suggested, we have removed the mention of selection bias and its potential impact on the generalizability of our findings from the methods section. This information has now been properly included in the limitations section of the manuscript.

2. Reviewer says: Line 114 and 115, authors have mentioned that “they excluded people who had a doctor's diagnosis of depression, or who had been treated with antidepressants in the previous year. How about anxiety and other clinical conditions, why only excluding depression. As we know that during pandemic more anxiety was reported then depression, kindly clarify this.

Our response: Thank you for pointing out this important detail. We apologize for the error in the initial version of the manuscript. In fact, both depression and anxiety were considered exclusion criteria for participation in the study due to their potential impact on the participants' mental health status and the risk of bias in the study outcomes. This has now been clarified and reformulated in lines 115-119 of the methods section.

3. Reviewer says: Authors are requested to give description of scales/instruments in subsection separately. They are requested to mention the operational definitions separately at the end of the method section, it should not be mixed up with the description of scales/instruments.

Our response: We appreciate the reviewer’s observation. As requested, we have reorganized the information in the methods section to clearly separate the description of the scales from the operational definitions. The description of each instrument (Insomnia Severity Index, Smartphone Dependence and Addiction Scale, PHQ-9, and GAD-7) is now located in a separate subsection, while the operational definitions of the variables of interest (insomnia, depressive and anxiety symptoms) have been placed at the end of the methods section, as requested.

Reviewer #3:

1. Reviewer says: 1. In Ethics statement, please "indicate the form of consent obtained"

2. 2. Please find the comments for given line numbers below:

Our response: Thank you for your valuable comment. We have added more detailed information regarding the form of consent obtained as follows: "All participants provided virtual informed consent before proceeding with the survey. This was achieved through a consent form embedded at the beginning of the online questionnaire. Participants were required to read and agree to the consent terms before accessing the survey questions. (lines 398-416).

3. Reviewer says: Line 95. How many universities were selected? Were all universities in Peru included? If not, what sampling strategy was used?

Our response: Thank you for your comment. Three universities were selected for this study, all of which offer the human medicine program in the Piura region, one of the most affected areas in terms of COVID-19 morbidity and mortality in Peru. These universities were selected based on their availability and willingness to participate, and no specific sampling strategy was used to select them.

The sampling approach used for this study was non-probabilistic and based on convenience, which was the most feasible option given the logistical constraints and limitations posed by the pandemic. We acknowledge the potential limitations of this approach, including the possibility of selection bias, which may affect the generalizability of the findings. These limitations are discussed further in the results and discussion sections. This has been included in lines 173-207 of the Methods section.

4. Reviewer says: Line 116. How many participants were included in the survey? And how many were excluded to reach the final sample size of 370?

Our response: The survey included a total of 370 participants, representing a subset of the medical student population across three universities. Of the initial participants, 38 were excluded due to a history of depression/anxiety, either diagnosed by a healthcare professional or having been treated with antidepressants or anxiolytics in the previous year. The final sample size of 370 participants was reached after applying this exclusion criterion. This information is discussed in detail between lines 198-207 of the Methods section.

5. Reviewer says: Line 157. I suppose, 15-19 should be categorized as 'moderately severe' instead of 'moderate'.

Our response: Thank you for your suggestion. We have re

---

## [Decision Letter · Decision Letter 1]

Dear Dr. Valladares-Garrido,

Thank you for submitting your manuscript to PLOS ONE. After careful consideration, we feel that it has merit but does not fully meet PLOS ONE’s publication criteria as it currently stands. Therefore, we invite you to submit a revised version of the manuscript that addresses the points raised during the review process.

The authors used both the terms dependence and addiction in the text. Somewhere they are distinct and somewhere they are interchanged. I would suggest to change this to avoid confusion and to use only one term or to clearly explain the difference between then in the text.

The SDAS cutoff points are pooled from the sample so they are sample dependent. They don't give absolute numbers which would be more sample independent. I would like to see more explanation abut this. With that kind of cutoff points any sample would produce both addictive and non-addictive participants.

The authors wrote `We found that smartphone dependence increased the prevalence of insomnia by 44.2%.` This result is from chi square test of association, so we can not draw conclusion what is the cause. Maybe it was the other way round or some third factor influencing both of those. The authors should change the language used to describe this result.

Sincerely,

Academic Editor

We look forward to receiving your revised manuscript.

Kind regards,

Pavle Randjelovic, Ph.D.

Academic Editor

PLOS ONE

Journal Requirements:

Reviewers' comments:

Reviewer's Responses to Questions

**Comments to the Author**

Reviewer #4: All comments have been addressed

2. Is the manuscript technically sound, and do the data support the conclusions?

Reviewer #4: Yes

3. Has the statistical analysis been performed appropriately and rigorously?

Reviewer #4: Yes

4. Have the authors made all data underlying the findings in their manuscript fully available?

Reviewer #4: Yes

5. Is the manuscript presented in an intelligible fashion and written in standard English?

Reviewer #4: Yes

Reviewer #4: Thank you for the opportunity to review the manuscript entitled "Smartphone dependence, addiction, and insomnia among medical students during the COVID-19 pandemic" (Manuscript ID: PONE-D-24-56765). The manuscript addresses an important and timely topic, particularly given the global reliance on digital devices and the mental health challenges exacerbated by the COVID-19 pandemic.

Overall Assessment:

The study is well-structured, the statistical analysis is appropriate, and the discussion is grounded in relevant literature. The use of validated instruments (ISI, PHQ-9, GAD-7, and SDAS) strengthens the methodological rigor, and the findings contribute meaningfully to the existing knowledge on smartphone dependence and sleep health in medical student populations.

Strengths:

The large sample size (n = 370) provides adequate power for the analyses.

Adjustments for potential confounders such as depression and anxiety enhance the internal validity of the results.

The study uses a well-defined and operationalized construct of smartphone dependence and addiction using a validated scale adapted for Spanish-speaking populations.

Areas for Improvement:

Clarity on Secondary Analysis:

While the authors mention this is a secondary analysis of previously collected data (ref [21]), greater clarity should be provided regarding the original study’s objectives, overlap in variables analyzed, and the novelty of this manuscript’s specific focus. This is essential to fully assess the extent of original contribution and avoid concerns of salami publication or dual submission.

Sampling and Generalizability:

The convenience sampling strategy and use of WhatsApp groups may limit generalizability. While this is acknowledged in the discussion, it may be helpful to provide more context or comparison with national student demographics to assess representativeness.

Data Availability:

The manuscript states “All relevant data are within the manuscript and its Supporting Information files.” However, it would be helpful to explicitly indicate where the datasets can be accessed or whether any anonymized data files are provided for replication purposes.

Ethical Considerations:

The authors adequately describe IRB approval and informed consent. However, it would strengthen the ethical narrative to specify whether participants were compensated (monetarily or through academic credits) and confirm that participation did not influence course evaluations or academic standing.

Language and Formatting:

A few sections would benefit from minor editorial revisions for clarity and flow, particularly in the Methods and Discussion. For instance, "not single" could be more clearly described as "married or in a relationship."

Future Directions:

While the authors suggest longitudinal and comparative studies in the future, it may be valuable to highlight the potential of interventions (e.g., smartphone use reduction strategies or sleep hygiene programs) that could stem from this evidence.

Ethics and Integrity:

There are no apparent concerns related to research misconduct or unethical practices. The ethics approval is clearly stated (CEI-EPM-UCV-2020), and there is no indication of duplicate publication at this time. However, the authors should ensure that the current manuscript provides distinct contributions relative to the prior publication (ref [21]).

**Do you want your identity to be public for this peer review?** For information about this choice, including consent withdrawal, please see our Privacy Policy

Reviewer #4: **Yes: ** Carlos Miguel Rios-Gonzalez

---

## [Author Response · Author response to Decision Letter 2]

11 Jun 2025

Dear Editor,

Thank you very much for reviewing our article, " Smartphone dependence, addiction, and insomnia among medical students during the COVID-19 pandemic ". Your suggestions and comments will be addressed below. Thank you for your valuable time and excellent review.

Editor's comments

1. Editor says: The authors used both the terms dependence and addiction in the text. Somewhere they are distinct and somewhere they are interchanged. I would suggest to change this to avoid confusion and to use only one term or to clearly explain the difference between then in the text.

Our response: Thank you for your valuable observation regarding the inconsistent use of the terms dependence and addiction throughout the manuscript. We acknowledge that in the previous version, these terms were sometimes used interchangeably and without sufficient conceptual clarification, which could lead to confusion for readers.

In response, we have carefully revised the manuscript to ensure precise and consistent terminology aligned with the structure of the Smartphone Dependence and Addiction Scale (SDAS), the instrument used in this study. This scale classifies smartphone use into three categories—nondependence, dependence, and addiction—based on increasing severity. To clarify this, we have added a concise conceptual distinction in the Materials and Methods section, stating that dependence refers to problematic but still functional smartphone use, whereas addiction denotes a more severe level, marked by loss of control and functional impairment. This clarification has also been echoed in the Introduction, the Abstract, and the Discussion to maintain internal consistency.

Moreover, in the Results section and tables, we have preserved both categories to respect the original scoring structure of the SDAS, using fixed cutoff values (≤2.23, 2.24–3.05, ≥3.06) established during the original validation study. A corresponding explanatory note has also been added to Table 3 to reflect this, and all prior references to sample-based percentiles have been corrected.

2. Editor says: The SDAS cutoff points are pooled from the sample so they are sample dependent. They don't give absolute numbers which would be more sample independent. I would like to see more explanation abut this. With that kind of cutoff points any sample would produce both addictive and non-addictive participants.

Our response: Thank you for the opportunity to clarify this methodological point. We understand the concern regarding the use of percentile-based cutoffs, which would indeed render the classification of smartphone use levels sample-dependent and limit generalizability. However, we would like to clarify that we did not generate cutoffs from our own sample distribution. Instead, we followed the original scoring criteria proposed by Aranda-López et al. (2017) in the development and validation of the “Escala de Dependencia y Adicción al Smartphone (EDAS)”. This validated scale establishes fixed thresholds based on average item scores to classify users into three categories:

• Nondependent use: mean score ≤ 2.23

• Dependent use: mean score between 2.24 and 3.05

• Addictive use: mean score ≥ 3.06

These thresholds were applied directly in our research and therefore do not vary across samples. The categorization in our study is sample-independent and consistent with the instrument’s validated structure. To avoid further confusion, we have revised the Materials and Methods section to explicitly state that the classification was based on predefined, validated cutoff scores from the original scale, and not on percentiles derived from our study sample.

3. Editor says: The authors wrote `We found that smartphone dependence increased the prevalence of insomnia by 44.2%.` This result is from chi square test of association, so we can not draw conclusion what is the cause. Maybe it was the other way round or some third factor influencing both of those. The authors should change the language used to describe this result.

Our response: Thank you for the valuable observation. We have revised the language in the Abstract, Results, and Discussion to ensure that only associations are reported, avoiding any implication of causality. The revised phrasing reflects that smartphone dependence and addiction were associated with higher prevalence of insomnia. We also added a statement in the Discussion and Conclusions acknowledging that the cross-sectional design does not permit causal inference.

4. Editor says: The same applies to `depression increased the prevalence of insomnia`. It is test of association.

Our response: Thank you for the observation. We have revised the language to reflect that depression was associated with a higher prevalence of insomnia, rather than suggesting a causal relationship. This change has been made in the Results and Discussion sections to ensure consistency with the cross-sectional nature of the study.

Reviewer #4:

1. Reviewer says: Thank you for the opportunity to review the manuscript entitled "Smartphone dependence, addiction, and insomnia among medical students during the COVID-19 pandemic" (Manuscript ID: PONE-D-24-56765). The manuscript addresses an important and timely topic, particularly given the global reliance on digital devices and the mental health challenges exacerbated by the COVID-19 pandemic.

Our response: Thank you for your positive feedback and for recognizing the relevance of our study. We appreciate your comments and have addressed them carefully to improve the manuscript.

2. Reviewer says: The study is well-structured, the statistical analysis is appropriate, and the discussion is grounded in relevant literature. The use of validated instruments (ISI, PHQ-9, GAD-7, and SDAS) strengthens the methodological rigor, and the findings contribute meaningfully to the existing knowledge on smartphone dependence and sleep health in medical student populations.

Our response: Thank you for your encouraging remarks. We appreciate your recognition of the study’s structure, methodological rigor, and contribution to the literature.

3. Reviewer says: The large sample size (n = 370) provides adequate power for the analyses. Adjustments for potential confounders such as depression and anxiety enhance the internal validity of the results.

Our response: Thank you for highlighting the strengths of our study design. We appreciate your recognition of the sample size and the inclusion of relevant covariates to improve internal validity.

4. Reviewer says: The study uses a well-defined and operationalized construct of smartphone dependence and addiction using a validated scale adapted for Spanish-speaking populations.

Our response: Thank you for acknowledging the use of a validated and culturally adapted scale. We agree that employing a well-defined construct enhances the robustness and relevance of our findings.

5. Reviewer says: Clarity on Secondary Analysis:

While the authors mention this is a secondary analysis of previously collected data (ref [21]), greater clarity should be provided regarding the original study’s objectives, overlap in variables analyzed, and the novelty of this manuscript’s specific focus. This is essential to fully assess the extent of original contribution and avoid concerns of salami publication or dual submission.

Our response: Thank you for this important observation. We confirm that the current manuscript is a secondary analysis of data originally collected for the study published as reference [21] (Santander-Hernández et al., PLOS ONE, 2022). In that study, the primary objective was to assess the association between smartphone overuse and symptoms of depression and anxiety. In contrast, the present manuscript is based on a different research question, focusing on the role of smartphone dependence and addiction—as defined by validated SDAS cutoffs—as the main exposure, and insomnia as the primary outcome. These variables (exposure and outcome) were not analyzed or reported in the original publication. Therefore, the analytic model, outcome of interest, and conceptual framing are all distinct, providing an original contribution to literature. We have now revised the Methods section to clearly state the objective of the primary study, the overlap in instruments, and the novelty of the present focus. We also confirm that there is no duplication or overlap in results, and that this manuscript addresses a unique and independent line of inquiry.

6. Reviewer says: Sampling and Generalizability:

The convenience sampling strategy and use of WhatsApp groups may limit generalizability. While this is acknowledged in the discussion, it may be helpful to provide more context or comparison with national student demographics to assess representativeness.

Our response: Thank you for this insightful suggestion. We acknowledge that the use of convenience sampling through WhatsApp groups may limit the generalizability of our findings. However, as noted in the revised Discussion section, recent multicenter mental health studies conducted in Peru among medical students during the COVID-19 pandemic have reported demographic characteristics—particularly in terms of age and sex—similar to those observed in our sample [PMID: 35201957, 35967544, 37923417, 37363900]. This comparability supports the representativeness of our sample and adds contextual value to the interpretation and applicability of our results despite the non-random sampling method.

7. Reviewer says: Data Availability:

The manuscript states “All relevant data are within the manuscript and its Supporting Information files.” However, it would be helpful to explicitly indicate where the datasets can be accessed or whether any anonymized data files are provided for replication purposes.

Our response: Thank you for this helpful suggestion. We have updated the Data Availability Statement to explicitly clarify that the anonymized dataset used for this analysis is included as a Supporting Information file (S1 Table. Database.), allowing full replication of the study. All relevant variables and coding procedures are documented for transparency and reproducibility.

8. Reviewer says: Ethical Considerations: The authors adequately describe IRB approval and informed consent. However, it would strengthen the ethical narrative to specify whether participants were compensated (monetarily or through academic credits) and confirm that participation did not influence course evaluations or academic standing.

Our response: Thank you for this important observation. We confirm that participants were not offered any monetary compensation or academic credit for their participation. Additionally, we clarify that participation was entirely voluntary and had no influence on course evaluations, grades, or academic standing. This information has been added to the Ethics Statement in the Materials and Methods section to strengthen the ethical transparency of the study.

9. Reviewer says: Language and Formatting:

A few sections would benefit from minor editorial revisions for clarity and flow, particularly in the Methods and Discussion. For instance, "not single" could be more clearly described as "married or in a relationship."

Our response: Thank you for your helpful observation. We have reviewed the manuscript for clarity and language consistency, particularly in the Methods and Discussion sections. The phrase “not single” has been revised to “married or in a relationship,” and other minor editorial improvements have been made throughout the manuscript to enhance readability and flow.

10. Reviewer says: Future Directions:

While the authors suggest longitudinal and comparative studies in the future, it may be valuable to highlight the potential of interventions (e.g., smartphone use reduction strategies or sleep hygiene programs) that could stem from this evidence.

Our response: Thank you for this thoughtful recommendation. In response, we have expanded the Discussion section to emphasize the practical relevance of our findings. Specifically, we now highlight the potential for developing and evaluating structured interventions—such as digital well-being education, sleep hygiene programs, and cognitive-behavioral strategies—that aim to mitigate insomnia and support mental health among medical students. This addition aligns with the observed associations and reinforces the translational value of our results.

11. Reviewer says: Ethics and Integrity:

There are no apparent concerns related to research misconduct or unethical practices. The ethics approval is clearly stated (CEI-EPM-UCV-2020), and there is no indication of duplicate publication at this time. However, the authors should ensure that the current manuscript provides distinct contributions relative to the prior publication (ref [21]).

Our response: Our response: Thank you for this observation. We confirm that this manuscript is a secondary analysis of data previously collected for a different research objective, as outlined in reference [21]. The present study addresses a distinct research question by examining the association between smartphone dependence/addiction and insomnia—variables not analyzed or reported in the prior publication. We have clarified this distinction in the Study Design subsection of the Materials and Methods section, ensuring full transparency and reinforcing the originality of the current contribution. There is no duplication of results or overlap in conclusions.

If you have any comments or recommendations, we are ready to respond.

Sincerely,

Mario J. Valladares-Garrido

Escuela de Medicina Humana, Universidad Señor de Sipán, Chiclayo, Peru; vgarrido@uss.edu.pe

---

## [Decision Letter · Decision Letter 2]

Smartphone dependence, addiction, and insomnia among medical students during the COVID-19 pandemic

PONE-D-24-56765R2

Dear Dr. Valladares-Garrido,

We’re pleased to inform you that your manuscript has been judged scientifically suitable for publication and will be formally accepted for publication once it meets all outstanding technical requirements.

Kind regards,

Pavle Randjelovic, Ph.D.

Academic Editor

PLOS ONE

Additional Editor Comments (optional):

Reviewers' comments:

Reviewer's Responses to Questions

**Comments to the Author**

Reviewer #4: All comments have been addressed

2. Is the manuscript technically sound, and do the data support the conclusions?

Reviewer #4: Yes

3. Has the statistical analysis been performed appropriately and rigorously?

Reviewer #4: Yes

4. Have the authors made all data underlying the findings in their manuscript fully available?

Reviewer #4: Yes

5. Is the manuscript presented in an intelligible fashion and written in standard English?

Reviewer #4: Yes

Reviewer #4: The manuscript now meets the requirements for publication, as all reviewer comments have been thoroughly addressed

**Do you want your identity to be public for this peer review?** For information about this choice, including consent withdrawal, please see our Privacy Policy

Reviewer #4: **Yes: ** Carlos Miguel Rios-Gonzalez

---

## [Editor Report · Acceptance letter]

PONE-D-24-56765R2

PLOS ONE

Dear Dr. Valladares-Garrido,

I'm pleased to inform you that your manuscript has been deemed suitable for publication in PLOS ONE. Congratulations! Your manuscript is now being handed over to our production team.

Kind regards,

on behalf of

Dr. Pavle Randjelovic

Academic Editor

PLOS ONE